# Artificial Neural Network for Predicting the Safe Temporary Artery Occlusion Time in Intracranial Aneurysmal Surgery

**DOI:** 10.3390/jcm10071464

**Published:** 2021-04-02

**Authors:** Shima Shahjouei, Seyed Mohammad Ghodsi, Morteza Zangeneh Soroush, Saeed Ansari, Shahab Kamali-Ardakani

**Affiliations:** 1Neurology Department, Neuroscience Institute, Geisinger Health System, Danville, PA 17822, USA; 2Department of Neurosurgery, Tehran University of Medical Sciences, Tehran 14155-6559, Iran; ghodsism@sina.tums.ac.ir (S.M.G.); kamalishahab30@yahoo.com (S.K.-A.); 3Bio-Intelligence Research Unit, Electrical Engeneering Department, Sharif University of Technology, Tehran 14588-89694, Iran; morteza.soroush@sharif.edu; 4Department of Biomedical Engineering, Science and Research Branch, Islamic Azad University, Tehran 14778-93855, Iran; 5National Institute of Neurological Disorders and Stroke, National Institute of Health, Bethesda, MD 20892, USA; saeed.ansarisadrabadi@nih.gov

**Keywords:** aneurysm surgery, temporary artery occlusion, clipping time, artificial neural network

## Abstract

Background. Temporary artery clipping facilitates safe cerebral aneurysm management, besides a risk for cerebral ischemia. We developed an artificial neural network (ANN) to predict the safe clipping time of temporary artery occlusion (TAO) during intracranial aneurysm surgery. Method. We devised a three-layer model to predict the safe clipping time for TAO. We considered age, the diameter of the right and left middle cerebral arteries (MCAs), the diameter of the right and left A1 segment of anterior cerebral arteries (ACAs), the diameter of the anterior communicating artery, mean velocity of flow at the right and left MCAs, and the mean velocity of flow at the right and left ACAs, as well as the Fisher grading scale of brain CT scans as the input values for the model. Results. This study included 125 patients: 105 patients from a retrospective cohort for training the model and 20 patients from a prospective cohort for validating the model. The output of the neural network yielded up to 960 s overall safe clipping time for TAO. The input values with the greatest impact on safe TAO were mean velocity of blood at left MCA and left ACA, and Fisher grading scale of brain CT scan. Conclusion. This study presents an axillary framework to improve the accuracy of the estimated safe clipping time interval of temporary artery occlusion in intracranial aneurysm surgery.

## 1. Introduction

Intracranial aneurysms have a prevalence of 3.2% in the general population [1]. Although the majority of patients can remain asymptomatic, cerebral aneurysms have a significant risk of rupture. Temporary artery occlusion (TAO) is an indispensable technique to facilitate aneurysm dissection and clipping and to reduce the risk of intra-operative rupture [2]. However, TAO may be complicated with detrimental consequences such as cerebral ischemia and postoperative neurological deficits [3]. Thereby, estimating a safe clipping time (SCT) for TAO is essential to give the surgeons the maximum window to perform the surgery, and keep the patients safe from the complications of the surgery. Although several intra-operative neurophysiologic monitoring and imaging methods have been proposed for determining safe occlusion time [4,5], SCT is mostly estimated based on clinicians’ expertise in real practice. The purpose of this study is to leverage machine learning to identify the prominent clinical features determining the outcome of TAO and to predict the SCT for intracranial aneurysm surgeries.

Machine learning can be used to extract meaningful relationships and patterns from a set of features (model inputs) for estimating the future values of a phenomenon (model outcome). An artificial neural network (ANN) is a type of data mining and pattern recognition method which reveals complex nonlinear relationships in addition to linear correlations. ANNs have been widely used in a variety of neurosurgical applications, such as predicting the occurrence of symptomatic cerebral vasospasm after aneurysmal subarachnoid hemorrhage [6], traumatic brain injury outcome and survival [7,8], recurrent lumbar disk herniation [9], and endoscopic third ventriculostomy success in childhood hydrocephalus [10]. Regarding cerebral aneurysm surgeries, the majority of the studies deployed these techniques to predict the aneurysm rupture [11,12], or for automated detection of the aneurysms on imaging [13]. In this study, we aimed to evaluate the feasibility and validity of ANN modeling in predicting the SCT and for determining the prominent clinical features of cerebral aneurysm surgeries.

## 2. Methods

This study was conducted in Shariati Hospital, Tehran, Iran. To develop the ANN, we used two separate datasets.

### 2.1. Retrospective Cohort, Training, and Testing Set of the Model

We retrospectively reviewed the medical records of all patients who underwent craniotomy and clipping for aneurysm management between 2004 and 2011. Clinical data, including demographic information, comorbidities, pre- and post-neurological examination, Fisher grading scale of computerized tomography (CT) scan imaging, pre- and post-operative trans cranial doppler (TCD), location and diameter of the aneurysm(s), and temporary artery clipping time and number(s), were extracted. The presence or absence of flow-through vessels of the circle of Willis and possible anatomic variations were indicated by either digital subtraction angiography (DSA), computed tomography angiography (CTA), magnetic resonance angiography (MRA), or T2 weighted magnetic resonance imaging (MRI). The mean velocity of flow in cerebral arteries was measured from pre-operative TCD.

The information from the patients in the retrospective cohort was used to train the model. To obtain the SCT, we excluded all patients with unfavorable outcomes or any signs of ischemia. Patients with Glasgow coma scale (GCS) less than 11, presence of a neurologic deficit in the pre-operative examination, post-operative decline in either motor or sensory function, or any pathologic finding in neuroimaging other than the presence of aneurysm were excluded from the training set.

### 2.2. Prospective Cohort and Validation Set of the Model

Between 2011 and 2013, we devised a protocol to prospectively include the patients with surgical clipping of cerebral aneurysms (ruptured and un-ruptured). We only included those with aneurysms of the anterior communicating artery (AcomA) or middle cerebral arteries (MCA). Data were collected using the same protocol as the retrospective cohort. In addition, all patients of the prospective cohort underwent diffusion weighted imaging (DWI) MRI within 6 h and 24 h of the surgery to rule out cerebral ischemia. We also measured the diameter of arteries in the circle of Willis (anterior cerebral arteries (ACA), AcomA, and MCA) from CTA images. Image-J software (Image J 1.42q software, U.S. National Institutes of Health, Bethesda, MA, USA) was used for this purpose. The information obtained from the prospective cohort was used to test and validate the model.

### 2.3. Surgical Techniques

The surgical procedure for clipping of the aneurysms was either a standard pterional craniotomy (MCA location) or frontotemporal craniotomy (AcomA location). For AcomA aneurysms, the ipsilateral and contralateral A1 segments were exposed and temporarily clipped. When the AcomA segment aneurysm was dissected and permanently clipped, the temporal clipping of the A1 segments was subsequently removed. In MCA aneurysms, the MCA was exposed from proximal to distal to identify the location of the aneurysm. Temporal clipping of the proximal MCA at the M1 segment was applied and then the aneurysm was dissected. Subsequently, the temporal clip of the MCA was removed. The duration of temporary vascular obstruction following clipping was measured in seconds.

### 2.4. Feature Selection for ANN Model

Through a comprehensive literature review and consultation with clinicians, a wide variety of related clinical and physiological parameters with a possible impact on the SCT were proposed. Based on the anatomical distribution of intracranial aneurysms and the importance of compensatory blood flow mechanisms in each segment of the circle of Willis, 11 features were selected as the input for the ANN model.

Age, the diameter of the right and left MCAs, the diameter of the right and left A1 segment of ACAs, the diameter of AcomA, mean velocity of flow at the right and left MCAs, mean velocity of flow at the right and left ACAs, and Fisher grading scale of brain CT scan were considered as the input values for the model. The diameter of the P1 segment of the right and left posterior cerebral arteries (PCAs) and flow in the posterior circulation were excluded in our final model due to the low prevalence of posterior aneurysms.

### 2.5. Structure of the ANN Model

A three-layer structure neural network was used in this study: an input layer, one hidden layer, and an output layer (Figure 1). The number of input values (units) in the first layer of the model was equal to 11, the same as the number of selected features that was proposed to affect the outcome of clipping and subsequent ischemia. For determining the number and structure of the hidden layer, we considered the training accuracy and generalization. The presence of too many hidden layers (which is needed for accuracy) may cause overtraining, and this will result in a decline in generalization. To apply the optimum number of neurons in the hidden layer, the model was run with different counts. The architecture with five units on the hidden layer was accompanied by the lowest error. The output layer consisted of only one neuron, representing the SCT as the outcome of the model.

The units within each layer of the model were connected with the units of the adjacent layers through directed edges (weights). There were no connections between the units within the same layer [14]. A nonlinear Sigmoid function was applied to the hidden layer, and a linear function was applied to the output layer.

### 2.6. Training and Validating of the ANN Model

Five-fold cross-validation was used for this model. In each run of the modeling, 80% of the retrospective cohort was randomly selected to train the ANN model. The remaining 20% of the dataset was used to test the performance of the model. During the training phase, the weights and interactions of the input variables were gradually determined during each run. For this purpose, each set of input features was broadcast to every unit in the hidden layer. After computing its activation, each unit in the hidden layer transferred the signal to the unit of the adjacent output layer. In this way, the response of the network was computed for a specific set of input values (feed-forward propagation phase). In the backward propagation phase, the computed activation in the output layer (predicted SCT) was compared with the observed SCT value (obtained from the patient’s medical record), and the training error was calculated. The error was then propagated back to each unit of the hidden layer and updated the weights between the output layer and the hidden units. Correspondingly, the computed error in this layer was distributed to the input layer and the weights between the hidden layer and input layer were updated as well. This process was repeated several times, using different random sets of patients for training and testing the ANN model.

Data from the prospective cohort of the patients were used to validate the model and provide the performance metrics for the model. We used the trained ANN model (based on data from the retrospective cohort) to predict the SCT for patients in the prospective cohort. This cohort was kept unseen from the ANN algorithms in the training phase to prevent bias and overfitting.

### 2.7. Importance of Each Clinical Feature in Predicting the SCT

To evaluate the importance of the input parameters in predicting the SCT, a model sensitivity test was implemented. For this purpose, we considered a fixed weight of 1 for all the input variables. In each turn, we increased the weight of one variable up to 10% and evaluated the variation in the output value. After repeating this process for each input parameter, we ranked the obtained sensitivity values.

### 2.8. Estimating the Errors

To evaluate the proposed pattern recognition model performance, two types of error were proposed. The mean absolute deviation (Equation (1)) calculated the difference between the clinical assigned SCT values in real practice and those predicted by the model.
(1)$$Mean\;Absolute\;Deviation\;=\;\frac{\sum_{i=1}^{N_{tr}}\left|SCT_{i}-\hat{SCT_{i}}\right|}{N_{te}}\;\%$$
where *N* is the total number of patients, $N_{tr}=\frac{4N}{K}$ the total number of training samples, $N_{te}=\frac{N}{K}$ the total number of test samples, K=5 the realization of the K-fold validation algorithm in our model, SCT stands for safe clipping time, and $\hat{SCT_{i}}$ is the predicted value as the outcome of the model.

For relative error (Equation (2)), the mean absolute deviation was adjusted by the greatness of the error according to each value of the observed outcome. This criterion resulted in a better perception of the bias on the model. The relative error was considered to report the bias of the model in this study.
(2)$$Relative\;Error=\frac{1}{N_{te}}\sum_{i=1}^{N_{tr}}\left|\;\frac{SCT_{i}-\hat{SCT_{i}}}{SCT_{i}}\right|\;\%$$
where the parameters are as above.

MATLAB software was used for mathematical modeling and designing the ANN. The confidence interval of 95% was assigned to the outputs. We used a T-test to assess the independence of the outputs by considering a *p* value less than 5% as significant.

## 3. Results

A total of 131 patients were evaluated for this study (105 patients from the retrospective cohort and 26 patients from the prospective cohort). Six patients were excluded from the prospective cohort due to low GCS (4 patients) and positive DWI MRI indicating postoperative cerebral ischemia (2 patients, none could be directly related to temporary clipping). Demographic data of the included patients, location of the aneurysms, and also details of the Fisher grading scale of each cohort are available in Table 1.

The overall predicted TAO based on the prospective cohort was 90–960 s; 120–932 s in AcomA, 240–960 s in right MCA, and 90–950 s in left MCA (Table 2). The average deviation of predicted SCT by the ANN model in this study from the clinical observed SCT of the unseen prospective cohort was 12%, leaving an 88% accuracy of the model.

A sensitivity analysis of the input values showed that mean velocity of the left M1, mean velocity of the left A1, and Fisher grading scale had the greatest impact on SCT (Table 3).

## 4. Discussion

Surgical management of aneurysms is among the most critical procedures in neurosurgery. Temporary artery occlusion (TAO) is a fundamental component in facilitating aneurysm dissection. The main purpose of this study was to introduce an alternative intelligent predictive tool besides the commonly accepted clinical experience, rather than providing an absolute value for SCT. However, the ANN model in this study demonstrated that the clipping time might be considered as safe for intervals longer than those practiced in the clinic. We observed that mean velocity of flow at the left MCA and left ACA, in addition to the Fisher grading scale of brain CT scans have the greatest impact on the outcome of the TAO.

Although a detrimental consequence of clipping is ischemia, several mechanisms such as redirection of blood flow from the contralateral side through communicating arteries of the Willis circle, leptomeningeal and collateral vessels, and cortical anastomosis can compensate for the hypo-perfusion and eliminate the cerebral ischemia [15,16,17,18]. Aging has been shown to reduce the efficacy of collateral flow and cortical anastomosis capacity by decreasing the collateral number and diameter, increasing tortuosity, and impairment of remodeling capacity [19,20,21].

In addition, the difference in predicted SCT for different vessels might have a biological basis. Predicted SCTs were higher in the left hemisphere. The difference in the origin of right and left common carotid arteries (aortic arc versus the brachiocephalic artery on the right side), the curvature of the vessels, carotid intima-media thickness (CIMT), and other hemodynamic characteristics of the vessels in the right and left side may result in variation between flow in the right and left circulation [22,23]. Blood flow in each vascular section is a function of the velocity of blood and diameter of the vessel at that section (Flow = Velocity × Diameter). Accordingly, by considering the similar diameter of vessels on both sides, the higher velocity of the blood on the left side might be representative of the greater flow in the left circulation. This might be the underlying reason why the velocity of the left ACA and MCA has a major impact on the outcome. The higher incidence rate of aneurysm formation and greater wall shear stress (WSS) and wall shear stress gradient (WSSG) on the left side in comparison with the right in our study (not presented in this draft), may verify this assumption. Additionally, the difference in blood flow may produce a higher compensatory potential for the dominant side in case of vessel occlusion, by redirecting the blood flow through the Willis circle toward the site of obstruction. Consequently, the extra ten seconds of safe occlusion time in the right MCA TAO (960 s versus 950 s for left MCA TAO), although clinically insignificant, may demonstrate this bonus reperfusion provided by the contralateral dominant side.

### 4.1. Selected Features as Input of the Model

WSS and WSSG can affect the SCT by promoting aneurysm formation. Permanent pathologic alteration of vasculature, such as disruption of the internal elastic lamina or thinning of the media along with increasing the number and tortuosity of collateral vessels, were introduced as complications of WSS and WSSG [19,24,25,26,27]. Alteration in hemodynamic parameters such as the diameter of vessels and velocity of blood flow can change WSS and WSSG [28,29,30]. Thereby, we considered the diameter of ACAs, MCAs, and AcomA, and the mean velocity of ACAs and MCAs as an indirect measure of WSS and WSSG.

Primarily, we considered the diameter of the P1 segment of right and left PCA and flow in posterior circulation as other predictors of SCT. Previous studies suggested that AcomA is more prominent in maintaining the blood flow after obstruction than the posterior communicating arteries [25,31]. Besides this, aneurysms are not uniformly distributed [32]. Less than 1% of intracranial aneurysms occur at the vertebra–basilar junction, basilar artery, and superior cerebellar artery bifurcation. ACA and MCA bifurcations together account for more than 50% of intracranial aneurysms [33]. Consequently, we did not include the diameter of right and left P1 and flow in the posterior circulation in our final model.

### 4.2. Limitations and Error Estimation

The strength of this study was to include information of the patients from two different cohorts in model training, using cross-validation from a retrospective cohort, and model testing using a prospective cohort. Using a very select number of features with clinical value was important to ensure our study did not suffer from missingness, which could have introduced selection bias. We considered a comprehensive panel of clinical and imaging features as the input to assess the feasibility of our approach. However, the Institutional Review Board (IRB) prevented us from including sex as an input variable in the validation cohort due to the deidentification process for datasets including less common pathologies with fewer than 100 patients. Despite considering various imaging modalities to monitor the possible post-operative ischemia, determining the exact underlying cause of ischemia (e.g., impact of final clipping rather than temporary clipping, vasospasm, and other intra- or post-operative complications) was challenging, and we did not include patients with cerebral ischemia in our models. Adding intraoperative variables and patients with adverse outcome could improve the predictive value of our ANN model.

The average deviation of predicted SCT from clinically assigned SCT (relative error of our ANN) was equal to 12%. In the training phase, we used five-fold cross validation, which resulted in average relative regression errors of 4.3% (training set) and 11.3% (test set). This training error can be considered quite low and acceptable for training process. After finalizing our regression model, we employed our final model to estimate SCT for the validation set (prospective cohort). This result indicates that our model does not suffer from overfitting or underfitting or unequal distribution over different subsets. Although, an 88% accuracy is a promising result for our pilot study with a total of 131 patients, the model would benefit from validation and justification over larger datasets. However, considering the prevalence of cerebral aneurysms which need surgical intervention, including a large cohort of patients is not simple. Furthermore, in this pilot feasibility study, we used ANN as our machine learning framework, however; comparative analysis with other modeling tools and deep learning methods may provide better performance. We will employ commonly used regression models in our future study to better visualize the power of our model compared to previously used linear models in the medical literature.

## 5. Conclusions

The main goal of this study was to present an axillary framework to improve the accuracy of the estimated safe clipping time interval of temporary artery occlusion during intracranial aneurysm surgery. The proposed method was an offline approach that can provide a prediction for the SCT in TAO before the surgery. However, to provide an accurate and precise SCT during the surgery, integration of online measurements and frequent updates of the predicted clipping time is required. To design a model with higher generalization, further studies with more clinical variables, larger sample size, and more diverse demographics are recommended.

## Figures and Tables

**Figure 1 jcm-10-01464-f001:**
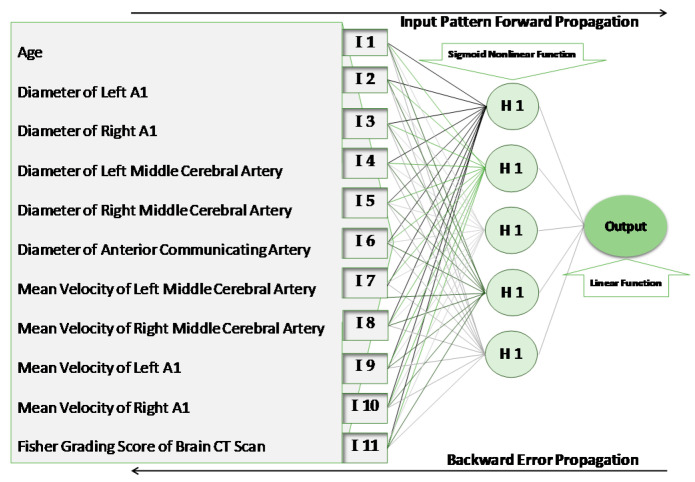
The structure of the artificial neural network. I, input unit; H, hidden unit.

**Table 1 jcm-10-01464-t001:** Demographic and surgical characteristics of the patients in retrospective and prospective datasets.

Parameter	Retrospective Dataset (*n* = 105)	Prospective Dataset (*n* = 20)
Age; Mean (Range); Years	48.7 (30–72)	50.2 (30–76)
Men; *n* (%)	32 (30.4%)	9 (45%)
Location of Aneurysm		
Middle Cerebral Artery; *n* (%)	80 (76.1%)	14 (70%)
Anterior Communicating Artery; *n* (%)	25 (23.9%)	6 (30%)
Fisher Grading Scale of CT Scan Images		
One; *n* (%)	28 (26.66%)	1 (5%)
Two; *n* (%)	46 (43.8%)	9 (45%)
Three; *n* (%)	23 (21.9%)	4 (20%)
Four; *n* (%)	8 (7.61%)	6 (30%)

**Table 2 jcm-10-01464-t002:** Output values. The safe clipping time interval is based on the aneurysm location.

Site of Obstruction	Safe Time Interval (Seconds)
Overall	90–960
Right Middle Cerebral Artery	240–960
Left Middle Cerebral Artery	90–950
Anterior Communicating Artery	120–932

**Table 3 jcm-10-01464-t003:** Ranked output of the sensitivity analysis.

Rank	Input Value	Sensitivity (%)
1	Mean velocity of flow at left MCA (middle cerebral arteries)	73.82 ± 1.95
2	Mean velocity of flow at left ACA (anterior cerebral arteries)	67.23 ± 2.74
3	Fisher grading scale of brain CT scan	65.71 ± 5.31
4	Mean velocity of flow at right ACA	63.87 ± 4.82
5	Diameter of right MCA	59.22 ± 5.24
6	Diameter of AcomA (anterior communicating artery)	57.56 ± 3.13
7	Diameter of left MCA	55.59 ± 3.13
8	Diameter of left A1	45.74 ± 2.47
9	Age	41.35 ± 1.78
10	Mean velocity of flow at right MCA	32.19 ± 3.62
11	Diameter of right A1	23.45 ± 2.15

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
