# Peer review of "Artificial Neural Network for Predicting the Safe Temporary Artery Occlusion Time in Intracranial Aneurysmal Surgery"

_jcm, 2021, doi:10.3390/jcm10071464_

Round 1

Reviewer 1 Report

Shahjouei et al. conducted a study in which they aimed to build an artificial neural network (ANN) to predict the safe temporary artery occlusion time in intracranial aneurysmal surgery. They have collected relevant clinical/imaging data from both a retrospective and a prospective cohort of modest size and trained 3-layer ANN (input, one hidden layer, output) for a regression task of predicting safe clipping time (SCT). Their final model had an accuracy of 88%, demonstrating good performance aligned with their goal. Manuscript is well-written and the topic is of great clinical interest and utility.

Comments:

  • ANNs are of greatest value when they are utilized for pattern recognition in high dimensional data, that is why they have been most successful with tasks such as image classification (e.g., treating every pixel as an input, each layer of a convolutional ANN starts to pick up subtle patterns that build up to more concrete patterns/classifications as you get closer to the final output layer). The power of these algorithms come from the fact that with enough data, they have the ability to be the universal estimator for any function that connects the input to the output of interest. I am curious as to why authors just did not use multivariable linear regression for this task? What was the rationale for using an ANN? Do authors expect a lot of non-linearities and higher order interactions within these 11 features that they cannot possibly capture with multivariable regression (even with spline transformations and multiple interactions within features). That might have been more efficient given the sample size limitations and the fact that ANNs would require larger sample sizes for model fitting in a situation like this.
  • Was there a specific reason ‘sex’ was not included as a feature in the model training? Age and sex are two features that almost universally associate with all medical conditions.
  • Authors mention the average deviation from observed SCT was 12% per 5-fold cross-validation, did they not evaluate the accuracy in the final prospective validation cohort? If they did, why is this not given in the manuscript.

Author Response

  • ANNs are of greatest value when they are utilized for pattern recognition in high dimensional data, that is why they have been most successful with tasks such as image classification (e.g., treating every pixel as an input, each layer of a convolutional ANN starts to pick up subtle patterns that build up to more concrete patterns/classifications as you get closer to the final output layer). The power of these algorithms come from the fact that with enough data, they have the ability to be the universal estimator for any function that connects the input to the output of interest. I am curious as to why authors just did not use multivariable linear regression for this task? What was the rationale for using an ANN? Do authors expect a lot of non-linearities and higher order interactions within these 11 features that they cannot possibly capture with multivariable regression (even with spline transformations and multiple interactions within features). That might have been more efficient given the sample size limitations and the fact that ANNs would require larger sample sizes for model fitting in a situation like this.

Response:

We appreciate your constructive comments. You are right. In cohorts with a large sample size and high dimensional data, MLP models proposed by ANNs are superior to linear regressions. There are a couple of methods that could address the question of this study with the current dataset in a simpler approach. However, as we have mentioned in page 6, lines 183-185, the main purpose of this study was to introduce an alternative intelligent predictive tool besides the commonly accepted clinical experience and assess its feasibility. We had employed several methods before we decided to use MLP. Unfortunately, those methods such as linear regression and k-nearest neighbor were not as effective as we expected. The reason (as you mentioned) might be the nonlinearity in our complex data. Multivariable regression was not that effective, as it views the problem from a statistical point of view, and the nonlinearity in our data results in non-stationary data. This is one reason that we believe some methods such as multivariable regression was not that successful in predicting the outcome. In our future study, we are going to consider a comprehensive panel of baseline clinical characteristics, imaging, and surgical data in a large sample size of over 15 years. Per the insight from your comment, we will employ the multivariable linear regression in addition to the MLP and demonstrate the possible difference among these methods. To address this comment, we edited the manuscript as follows:

Page 7; lines 246-250: Furthermore, in this pilot feasibility study, we used ANN as our machine learning framework, however; comparative analysis with other modeling tools and deep learning methods may provide better performance. We will employ commonly used regression models in our future study to better visualize the power of our model compared to previously used linear models in medical literature.  

  • Was there a specific reason ‘sex’ was not included as a feature in the model training? Age and sex are two features that almost universally associate with all medical conditions.

Response:

Thanks for your feedback. As you correctly mentioned, age and sex are among the two items that should be considered in almost every model. We had two reason for excluding sex from our model. The main reason was the whole purpose of the study is to provide insight regarding the impact of each variable on the outcome to enable us to modify the variable as much as we can and improve the outcome of surgery. Decision making on elective surgeries of aneurysmal surgery can be influenced by age of the patients, but sex is not something that we can modify. In addition, due to the sample size of less than 100 in prospective cohort (validation cohort) the institutional review board (IRB) prevented us from including both age and sex at the same time (a policy for assuring the deidentification process of less common pathologies). Hopefully in our next study we would be able to add sex as an input value. We address this concern as follows:

Page 7, lines 230-232: We considered a comprehensive panel of clinical and imaging features as the input to assess the feasibility of our approach. However, Institutional Review Board (IRB) prevented us to include sex as an input variable in the validation cohort due to deidentification process for datasets including less common pathologies with fewer than 100 patients.

  • Authors mention the average deviation from observed SCT was 12% per 5-fold cross-validation, did they not evaluate the accuracy in the final prospective validation cohort? If they did, why is this not given in the manuscript.

Response:

We appreciate the comment. Based on the insight from this constructive comment, we edited the manuscript as follows:

Page 7, lines 238-243: The average deviation of predicted SCT from clinical assigned SCT (relative error of our ANN) was equal to 12%. In the training phase, we used 5-fold cross validation which results in the average relative regression errors of 4.3% (training set) and 11.3% (test set). This training error can be considered quite low and acceptable for training process. After finalizing our regression model, we employed our final model to estimate SCT for the validation set (prospective cohort). This result indicates that our model does not suffer from overfitting or underfitting or unequal distribution over different subsets.

Reviewer 2 Report

This is a  well-organized study for determining the safe clipping time interval of temporary artery occlusion during intracranial aneurysm surgery. The data are acceptable and the analysis are reliable. For the accuracy of the conclusion, it is also reasonable to design a model with higher generalization power,  further studies with more clinical variables, larger sample size, and more diverse demographics are recommended.  

Author Response

This is a  well-organized study for determining the safe clipping time interval of temporary artery occlusion during intracranial aneurysm surgery. The data are acceptable and the analysis are reliable. For the accuracy of the conclusion, it is also reasonable to design a model with higher generalization power,  further studies with more clinical variables, larger sample size, and more diverse demographics are recommended.  

Response:

The authors would like to appreciate reviewer two for evaluating this draft, in purpose of publishing in Journal of Clinical Medicine. Authors are delighted that the manuscript meet the quality considered by the reviewer. Authors will consider a complementary study in near future to address the shortcomings of the current study, as mentioned by reviewer.

Reviewer 3 Report

Manuscript describes "Artificial Neural Network for Predicting the Safe Temporary Artery Occlusion Time in Intracranial Aneurysmal Surgery". 

A few comments that may improve the manuscript for wider audience.

How many of the patients had ischemia after and related to temporary clipping? Could you identify for certainty that ischemia was related to temporary clipping and not related to rupture/vasospasm (per-operative or post/operative?)

Was ischemia calculated in overall neural tissue from MRI, or was there calculated only from temporary clipped distal part? How about success of the final clipping of the aneurysm. Was there failure? => Ischemia related to unwanted final clipping rather than temporary clipping? 

The safetime for clipping varies 90-960 s and as authors mention, it is highly dependent on collaterals also. It is somehow unclear how did you end-up with the timeframe.

Author Response

  • How many of the patients had ischemia after and related to temporary clipping? Could you identify for certainty that ischemia was related to temporary clipping and not related to rupture/vasospasm (per-operative or post/operative?)

Response:

We appreciate the comment and questions. As reviewer has correctly mentioned, differentiating the underlying cause of ischemia is a crucial issue in the models of safe clipping time prediction. Although there are several techniques to monitor different pathologies that may result in ischemia (such as vasospasm), determining the exact cause of observed ischemia in real practice can be challenging. In this study, we have used different modalities of imaging to consider possible complications of the aneurysmal surgery that can result in ischemia, such as measuring the diameter of vessels and blood flow for monitoring vasospasm. However, to prevent the mentioned bias, authors decided to only include the patients with satisfactory outcomes in absence of ischemic events. In fact, instead of designing the ANN model to predict if the patient had clipping-related ischemia or not (that could be biased with other causes of ischemia), the authors proposed a model to predict the safe time interval. In other word, by developing the ANN model, we tried to see the impact of variation of each input value on the safe predicted time and compare it with the real observations. This approach enables us to determine the possible determining factors and a clue for safe time margins. In the future studies, we will add the patients with different ischemic complications and intra-operative variables. To address this constructive comment, we modified the manuscript as follows:

Page 7, lines 232-237: Despite considering various imaging modalities to monitor the possible post-operative ischemia, determining the exact underlying cause of ischemia (e.g. impact of final clipping rather than temporary clipping, vasospasm, and other intra- or post- operative complications) was challenging and we did not include patients with cerebral ischemia in our models. Adding intraoperative variables and patients with adverse outcome could improve the predictive value of our ANN model.

  • was ischemia calculated in overall neural tissue from MRI, or was there calculated only from temporary clipped distal part? How about success of the final clipping of the aneurysm. Was there failure? => Ischemia related to unwanted final clipping rather than temporary clipping? 

Response:

We appreciate the comment. Imaging studies included the overall neural tissue. As mentioned in the method section (page2, line 60-67), patients in retrospective cohort were evaluated for Fisher grading scale, pre- and post- operative Trans Cranial Doppler, the presence or absence of flow-through vessels of the circle of Willis and possible anatomic variations were indicated by either DSA, CTA, or MRA. In addition, patients in the prospective cohort, received additional post-operative DWI within 6 hours and 24 hours and digital measurement of diameter of arteries in the circle of Willis from CTA.

We observed positive DWI findings in two patients, neither as a consequence of clipping. However, as mentioned in the previous question, only patients with satisfactory outcomes and absence of ischemia were included in the model. Thereby differentiating between ischemia related to temporary versus final clipping was not indicated for the current pilot study. These and other items related intra- and post-operative variables will be added to our future study. We added this limitation to the study as follows:

Page 5, lines 166-168: Six patients were excluded from the prospective cohort due to low GCS (4 patients) and positive DWI MRI indicating postoperative cerebral ischemia (2 patients, none could be directly related to temporary clipping).

Page 7, lines 232-237: Despite considering various imaging modalities to monitor the possible post-operative ischemia, determining the exact underlying cause of ischemia (e.g. impact of final clipping rather than temporary clipping, vasospasm, and other intra- or post- operative complications) was challenging and we did not include patients with cerebral ischemia in our models. Adding intraoperative variables and patients with adverse outcome could improve the predictive value of our ANN model.

  • The safe time for clipping varies 90-960 s and as authors mention, it is highly dependent on collaterals also. It is somehow unclear how did you end-up with the timeframe.

Response:

We appreciate the valuable question. Classically, the safe clipping time or any other procedure in clinic is measured from clinical observations. This method has several limitations, including the small sample size of the patients and its impact on the outcomes, and clinical perception of the surgeon for continuing the clipping. In this study, we proposed ANN models to determine the interaction and weight of each input value on the outcome (safe time).  In the retrospective cohort (building the model), in each turn of modeling 80% of the patients were selected randomly to train the model. During this step, the ANN model considered a possible relationship/interaction between variables and assigned a weight for each variable.  In the same turn, the model was tested by the remaining 20% data and the predicted safe time was compared to real practice. The possible error in this step was back propagated to update the weights and interactions of the input values. This step had been repeated several times with different random subsets as train and test. This number of repeats provides enough variability for the model and prevents overfitting of the model for a specific dataset (the main limitations of traditional models). When we tested this model with the unseen prospective cohort data, the model proposed the safe time intervals based on the weights and interactions of input variables from the retrospective cohort. The authors edited the manuscript as bellow:

Page 4, lines 126-131: Five-fold cross-validation was used for this model. In each run of the modeling, 80% of the retrospective cohort was randomly selected to train the ANN model. The remaining 20% of the dataset was used to test the performance of the model. During the training phase, the weights and interactions of the input variables were gradually determined during each run. For this means, each set of the input feature was broadcasted to every unit in the hidden layer.

Page 4, lines 140-142: Data from the prospective cohort of the patients were used to validate the model and provide the performance metrics for the model. We used the trained ANN model (based on data from retrospective cohort) to predict the SCT for patients in prospective cohort. This cohort was kept unseen from the ANN algorithms in the training phase to prevent bias and overfitting.

 Page 5, lines 172-175: The overall predicted TAO based on the prospective cohort was 90-960 seconds; 120-932 seconds in AcomA, 240-960 seconds in right MCA, and 90-950 seconds in left MCA (Table 2). The average deviation of predicted SCT by the ANN model in this study from the clinical observed SCT of the unseen prospective cohort was 12%, leaving an 88% accuracy of the model.

Page 7, lines 239-242: In the training phase, we used 5-fold cross validation which results in the average relative regression errors of 4.3% (training set) and 11.3% (test set). This training error can be considered quite low and acceptable for training process. After finalizing our regression model, we employed our final model to estimate SCT for the validation set (prospective cohort).

Round 2

Reviewer 1 Report

Authors have satisfactorily addressed my concerns.